# Treatment of Macular Edema in Vascular Retinal Diseases: A 2021 Update

**DOI:** 10.3390/jcm10225300

**Published:** 2021-11-15

**Authors:** Andrzej Grzybowski, Agne Markeviciute, Reda Zemaitiene

**Affiliations:** 1Department of Ophthalmology, University of Warmia and Mazury, 10-561 Olsztyn, Poland; ae.grzybowski@gmail.com; 2Institute for Research in Ophthalmology, 60-836 Poznan, Poland; 3Department of Ophthalmology, Medical Academy, Lithuanian University of Health Sciences, 50161 Kaunas, Lithuania; markeviciutee.agne@gmail.com

**Keywords:** macular edema, diabetic macular edema, retinal vein occlusion, laser photocoagulation, anti-VEGF, intravitreal injections

## Abstract

Macular edema (ME) is associated with various conditions; however, the main causes of ME are retinal vein occlusion (RVO) and diabetes. Laser photocoagulation, formerly the gold standard for the treatment of ME, has been replaced by anti-vascular endothelial growth factor (anti-VEGF) intravitreal injections. Despite its efficiency, this treatment requires frequent injections to preserve the outcomes of anti-VEGF therapy, and as many patients do not sufficiently respond to the treatment, ME is typically a chronic condition that can lead to permanent visual impairment. Generalized recommendations for the treatment of ME are lacking, which highlights the importance of reviewing treatment approaches, including recent anti-VEGFs, intravitreal steroid implants, and subthreshold micropulse lasers. We reviewed relevant studies, emphasizing the articles published between 2019 and 2021 and using the following keywords: macular edema, diabetic macular edema, retinal vein occlusion, laser photocoagulation, anti-VEGF, and intravitreal injections. Our results revealed that a combination of different treatment methods may be beneficial in resistant cases. Additionally, artificial intelligence (AI) is likely to help select the best treatment option for patients in the near future.

## 1. Introduction

Macular edema (ME) is a disease characterized by the swelling of the macula due to the abnormal accumulation of fluid [1]. It is associated with increased macular thickness and significantly reduced visual acuity, and it may develop in various ocular conditions.

Postoperative cystoid macular edema (PCME) typically occurs after cataract surgery; however, it can occur after any ocular surgery [2]. The increased phacoemulsification energy and phacoemulsification time or postoperative pseudophakodonesis can significantly contribute to PCME development [3]. It is thought that topical prostaglandin analogs used for glaucoma treatment may also promote PCME [3,4].

Corticosteroid eyedrops are prescribed postoperatively by most cataract surgeons to prevent the formation of PCME [5]. Topical steroids, non-steroidal anti-inflammatory eye drops, and ocular steroid injections (sub-tenon or intravitreal) are the main treatment options for PCME [2].

ME is the most common cause of vision loss in patients with uveitis [6,7]. Although both regional and systemic steroids are considered effective treatments, other treatment options are available, including immunomodulatory agents and anti-vascular endothelial growth factor (VEGF) intravitreal injections [7,8].

Cystoid macular edema (CME) is observed in patients with various retinal pathologies. It is considered a complication in patients with retinitis pigmentosa (RP), whereas tractional CME is associated with the persistent attachment of the vitreous at the macular region [9,10].

However, in most eyes undergoing treatment of ME related to retinal vascular disease, it is diabetic macular edema (DME) and retinal vein occlusion (RVO) that are the driving forces.

ME affects approximately 7 million patients with diabetic retinopathy (DR) and 3 million patients with retinal vein occlusion (RVO) [11].

The role of inherited genetic polymorphisms in DME development and treatment response is still poorly understood; nevertheless, possible DME risk genes have been identified. Graham and colleagues did not find any significant genome-wide associations with DME risk; however, they identified the top-ranked single nucleotide polymorphism (SNP) for DME in rs1990145 on chromosome 2 [12]. A trend toward an association between DME and DR was detected in two SNPs: rs12267418, near *MALRD1* (*p* = 0.008), and rs16999051 in the diabetes gene *PCSK2* (*p* = 0.007) [12,13]. It is clear that there is a need for larger studies.

CME involves fluid accumulation in the outer plexiform layer of the retina due to abnormal perifoveal retinal capillary permeability, whereas DME is associated with the leakage of macular capillaries and is observed in patients suffering from diabetes [14]. ME is also associated with an increase in VEGF and interleukin 6, which induce vascular permeability and vasodilation [15].

Chronic ME leads to permanent visual impairment by altering the outer limiting membrane, affecting photoreceptor segments (outer nuclear layer thinning and outer segment atrophy), and disorganization of inner retinal layers [11].

ME treatment approaches have changed substantially in recent years. Although laser photocoagulation (LP) has long been the gold standard for the treatment of ME, it is being replaced by anti-VEGF intravitreal injections, which have been reported as a first-line treatment for both DME and ME due to RVO.

This paper reviews and analyzes recent approaches to ME treatment and discusses future directions and perspectives in this field.

## 2. Methodology

A search of the medical literature was performed in PubMed and Google Scholar up to April 2021. The following keywords were used in various combinations: macular edema, diabetic macular edema, retinal vein occlusion, Laser Photocoagulation, anti-VEGF, intravitreal injections, and uveitis. Only articles with English abstracts focusing on ME caused by retinal vascular diseases, including DME and ME due to RVO, were reviewed. Studies were critically reviewed to construct an overview and guidance for further searches and highlight the lack of generalized recommendations. Emphasis was placed on articles published between 2019 and 2021.

## 3. Results

Intravitreal ranibizumab and aflibercept are currently approved for ME treatment, whereas bevacizumab is used off-label, and conbercept is approved and used for DME treatment only in China [16]. Frequent injections are required to preserve the effects of anti-VEGF therapy, and this treatment is therefore associated with repeated risk, high costs and an increasing burden on ophthalmologists and their patients. Despite the reported efficacy of anti-VEGFs, many patients do not respond well to treatment. In addition, identifying which treatment regimen is optimal is a constant dilemma. The main advantage of treat-and-extend (T and E) over pro re nata (PRN) regimens is a reduction in the number of hospital visits and recurrences [17]. Elsebaey and colleagues compared T and E treatment regimen with the PRN regimen in patients with DME [18]. They concluded that an individualized T and E regimen has the potential to reduce the clinic burden and improve patient compliance while maintaining effectiveness and providing well-tolerated treatment for DME [18]. Similar results were reported by Kim et al.: the T and E regimen of aflibercept in DME maintained effectiveness in a 2-year follow-up and reduced the number of injections compared with fixed dosing regimens [17].

Intravitreal corticosteroid implants ensure sustained drug release for a specific period and reduce the number of injections needed compared with anti-VEGF treatment. Steroid implants were reported to be effective and safe both in DME and ME due to RVO; however, they are typically used as a second choice in cases resistant to anti-VEGF treatment. The intravitreal dexamethasone (DEX) implant is approved for the treatment of DME and ME due to RVO; in the EU, it is approved for use in patients with DME that responds poorly to other treatments and for those who are pseudophakic or ineligible for other therapies [19]. The fluocinolone acetonide (FA) implant is approved for the treatment of DME and is typically used in patients who previously received a course of corticosteroids and did not experience a significant increase in eye pressure [20]. Despite the efficacy of steroids, they may be associated with increased intraocular pressure (IOP) and cataract formation.

Resistance to anti-VEGFs and intravitreal steroids treatment methods highlights the need for alternative treatment options.

### 3.1. Diabetic Macular Edema

The main DME treatment options are intravitreal injections of anti-VEGF agents and intravitreal corticosteroid injections. Formerly, macular LP was the gold standard for DME treatment; however, it is now utilized as an additional treatment. The two most common techniques of LP in patients with DME are focal photocoagulation targeting focal lesions (e.g., leaking microaneurysms or ischemic areas on fluorescein angiography (FA) for focal DME cases) and the grid laser technique, in which the laser is applied to diffuse leakages or nonperfusion areas; the latter is recommended for diffuse or more severe forms of DME [21,22]. According to the European Society of Retina Specialists (EURETINA) guidelines published in 2017, the focal and grid laser techniques should be utilized for non-center involving DME [23]. The laser can reportedly be applied in the vasogenic subform of DME, which is clinically characterized by the presence of focally grouped microaneurysms (MA) and leaking capillaries [24]. The primary reason grid laser is not recommended further is because of retinal scarring; however, when targeting capillary microaneurysms, a focal laser is beneficial as a second-line treatment [24,25]. In addition, it can be considered as a combined treatment option to reduce the number of anti-VEGF injections. Paques and colleagues performed a pilot study and reported significantly reduced macular thickness and improved visual acuity after elective photocoagulation of capillary microaneurysms in patients with chronic macular edema and severe hard exudates due to diabetic retinopathy or RVO [26].

Most studies found anti-VEGFs to be superior to laser treatment in DME patients. The REFINE study was conducted in Chinese patients with DME who received intravitreal ranibizumab injections or LP [27]. The results revealed a significantly greater improvement in mean best-corrected visual acuity (BCVA) at month 12 with ranibizumab than with LP [27]. Singh and colleagues reported that BCVA improvement was significantly greater with aflibercept than with laser techniques and was not influenced by any baseline factors [28,29]. A subthreshold micropulse laser (SML) is a relatively new tissue-sparing laser technique; it avoids protein coagulation and prevents retinal scars, allowing the preservation of retinal anatomy and function [30].

SML helps improve or stabilize visual function and decrease macular thickness in DME [31]. Vujosevic and colleagues performed a study that evaluated the effectiveness of SML treatment in patients with DME [31]. They reported that 31 patients (83.8%) required retreatment (mean number of SML treatments over 12 months: 2.19 ± 0.7); however, no eyes needed any additional treatments (anti-VEGF, steroids, and/or conventional laser) [31]. Al-Barki et al. compared the outcomes between short-pulse continuous wavelength and infrared micropulse lasers in DME treatment [32]. The authors concluded that the infrared micropulse system improved functional outcomes in patients with DME, whereas the short-pulse system resulted in a greater temporary reduction in edema [32]. Gawęcki and colleagues performed a systematized review and proposed that combining the SML treatment with anti-VEGFs may require fewer intravitreal injections than anti-VEGF monotherapy with equally favorable functional and morphological results in the ME treatment. However, SML alone was not superior to intravitreal treatment alone or combined treatment [33]. The authors noted that the studies under review varied in treatment protocols and inclusion criteria [33]. Altinel and colleagues compared the efficacy and safety of SML and intravitreal bevacizumab (IVB) injection combined therapy with IVB monotherapy in DME treatment [34]. They concluded that fewer IVB injections were needed when laser treatment was added; however, a significant increase in BCVA was not achieved [34]. Similarly, Furashova et al. reported that patients treated with ranibizumab combined with additional laser treatment experienced greater visual improvement and required fewer ranibizumab injections compared with patients treated only with ranibizumab [35].

Valera-Cornejo et al. evaluated the effect of SML treatment in center-involved DME in previously untreated (naïve) patients and patients who did not respond to prior treatment [36]. No significant changes in BCVA were observed between the groups after 3 months [36]. The change in central macular thickness (CMT) at 3 months was statistically but not clinically significant in the treatment-naïve group only, and no adverse events were reported [36]. Passos et al. reported that SML treatment used alone was not as effective as it could be when combined with other treatments [37]. DME cases associated with subretinal fluid had the best anatomical response, whereas intraretinal edema responded poorly to laser monotherapy [37]. The authors concluded that SML might be used in a combination treatment for ME [37]. Other authors also suggest considering laser therapy as an additional treatment in combination with intravitreal injections [21].

Anti-VEGFs utilize different molecules to achieve their effect: aptamers (pegaptanib); antibodies to VEGF (bevacizumab); antibody fragments to VEGF (ranibizumab); and fusion proteins, which combine a receptor for VEGF with the constant region of a human immunoglobulin (aflibercept and conbercept) [28]. Bevacizumab, ranibizumab, and aflibercept are the most common anti-VEGFs, and many studies have not observed significant differences in outcomes between them [28,38]. However, it has been suggested that the choice of anti-VEGF can be guided by the untreated BCVA. When it is lower, aflibercept has been suggested as the drug of choice [28,29]. The remaining anti-VEGFs, including bevacizumab, ranibizumab, and aflibercept, provide similar functional outcomes when the baseline BCVA is higher [28]. Bressler and colleagues, however, reported that after six consecutive injections, more patients presented with persistent ME following bevacizumab treatment compared with ranibizumab and aflibercept [39]. On this basis, Haritoglou et al. suggested switching from bevacizumab to either aflibercept or ranibizumab if DME persists while using bevacizumab [40].

Zhou et al. evaluated the efficacy and safety of intravitreal conbercept for DME treatment [41]. Patients were treated with one to three consecutive monthly intravitreal conbercept (IVC) injections, followed by retreatment with conbercept or switch therapy with triamcinolone acetonide (TA) based on a 6-month observation of the effect of treatment [29]. Approximately one-third of the eyes (29 of 89 eyes involved in the study) received intravitreal triamcinolone acetonide (IVTA) injections at month 6 [41]. The results revealed that the mean BCVA and CMT were significantly improved at 1 and 3 months after IVC treatment in the IVC group, and they gradually improved at 9 months after IVTA treatments in the IVC plus IVTA group [41]. Five eyes exhibited aggravated cataracts at the last follow-up visit after IVTA injection, and this was associated with the final decline in BCVA [41]. Nonetheless, the authors concluded that conbercept is safe and efficient, and TA may be beneficial in cases that are refractory to anti-VEGF treatment [41]. A meta-analysis comparing the efficacies of conbercept and ranibizumab for DME treatment demonstrated that intravitreal conbercept was significantly superior to ranibizumab in reducing CMT; however, no significant difference in visual improvement was observed [42]. The effects and safety of conbercept and ranibizumab in DME treatment were also compared in a recent meta-analysis by Sun et al., and the results demonstrated that intravitreal injections of conbercept were superior to ranibizumab in both reducing central retinal thickness and improving BCVA [43].

Corticosteroids are typically used as an alternative therapy for eyes with an insufficient response to anti-VEGF treatment reducing inflammation, decreasing the disruption of the blood–retinal barrier, and interfering with retinal angiogenesis [44]. Although intravitreal steroids are not used as often as anti-VEGFs, they can significantly reduce DME, and some authors suggest them as an option for first-line treatment. The main steroids used for the treatment of DME are TA, dexamethasone (DEX), and FA, which differ in their duration of action [40]. Because of the short vitreous elimination half-life of the solubilized fraction of these steroids, an extended duration of action can be achieved by applying sustained release systems (implants) into the vitreous cavity [40]. After one intravitreal injection of TA, the treatment effect was maintained for up to 6 months [40]. However, TA elevates the risk of increased IOP, and it may be associated with the risk of pseudoendophthalmitis [45,46] and retinal toxicity [47,48,49]; thus, it is used less frequently than its alternatives [40]. Additionally, TA has not been approved for DME treatment [28]. Conversely, the DEX drug release injectable implant has higher recognition, with a pharmacological effect ranging between 4 and 6 months [40].

A first-line treatment algorithm and guidelines in center-involving DME have been suggested by Kodjikian et al. [50]. The authors included a slow-release 700 µg dexamethasone intravitreal implant as an option for first-line treatment in center-involving DME, together with three anti-VEGFs (bevacizumab, ranibizumab, and aflibercept). Augustin and colleagues reported a consensus by a group of retina experts indicating that if a patient does not exhibit a sufficient response after 3–6 months of anti-VEGF treatment (a visual acuity gain of <5 ETDRS letters or a reduction in the central retinal thickness of ≤20%), switching to the dexamethasone implant should be considered [51]. An implant may also be suitable in eyes with massive lipid exudates or as a first-line treatment in pseudophakic patients, patients unwilling or unable to comply with tight anti-VEGF injection intervals, or patients with known vascular diseases [51].

Intravitreal DEX implants were reported to be effective in cases that were refractory to anti-VEGF treatment. Castro-Navarro and colleagues reported that the intravitreal DEX implant was effective and safe in both previously treated and untreated patients with DME [52]. Additionally, the authors observed that 6 months after the injection of the DEX implant, patients without prior DME treatment gained significantly more letters than patients who were previously treated [52]. These results suggest the possibility of achieving better results with earlier DEX implantation. This agrees with the results of a study by Medina-Baena, which demonstrated that at month 12, naïve patients exhibited a greater improvement in BCVA from baseline and achieved this BCVA improvement significantly faster than previously treated patients [53]. Similar results were observed in a study by Iglicki et al. [54]. They found that over a follow-up of 24 months, the vision in DME eyes improved after treatment with DEX implants in eyes that were treatment-naïve and in eyes that were refractory to anti-VEGF treatment; however, a greater improvement was observed in naïve eyes [54].

Although most studies evaluate CMT as the target of anatomical outcomes, Altun and colleagues evaluated the subfoveal choroidal thickness (SFCT) in vitrectomized eyes of patients with DME after intravitreal DEX implants [55]. The authors reported a statistically significant thinning of the mean SFCT during the follow-up period after DEX implant injection in vitrectomized eyes with DME [55].

Hong et al. performed a retrospective study to evaluate the effect of intravitreal TA injections in patients who were refractory to anti-VEGF treatment [44]. The authors reported that the BCVA improved significantly, and CMT was significantly reduced after a single TA intravitreal injection [44]. In addition, poorer visual acuity (VA) before the injection was associated with visual gain 1 month after the treatment [44]. Elevated IOP was observed in 17.1% of eyes, and this was observed significantly more often after IVTA injections containing a preservative than after preservative-free injections [39].

A longer pharmacological effect lasting up to 3 years can be achieved with an intravitreal FA sustained-release non-biodegradable device, which is inserted into the vitreous cavity via a 25-gauge needle; it contains 0.19 mg of FA and has a release rate of 0.2 μg/day [11]. Augustin and colleagues performed a retrospective study to evaluate the results of DME treatment with FA implants [56]. They concluded that a single FA implant could maintain reduced CMT for up to 3 years [56]. Several more studies reported similar results, highlighting that FA has a favorable safety and effectiveness profile while reducing CMT and improving BCVA [57,58,59]. Notably, Coelho and colleagues reported that FA exhibited long-term effectiveness in vitrectomized DME eyes and sustained the effectiveness in DME eyes that did not respond to DEX therapy [60].

The correct time to switch therapy if patients do not respond to anti-VEGF treatment remains unclear. Gonzalez et al. performed a study and reported that in eyes with poor responses after three anti-VEGF injections, it may be beneficial to switch to other modes of therapy [61]. Baker and colleagues found that for patients with DME and excellent visual acuity (defined as 20/25 or better), observation appeared to be a non-inferior initial management strategy compared with intravitreal aflibercept or LP in terms of visual acuity outcomes after 2 years [62]. Likewise, it was reported that initial focal or grid laser significantly reduced the risk of requiring aflibercept injection during follow-up [62].

Martínez and colleagues evaluated the effect of early DEX implantation in eyes with DME that received three or fewer anti-VEGF injections before the switch as well as the effect of later implantation in patients who received six or more anti-VEGF injections before the switch [63]. They reported that an early switch to DEX in patients who did not adequately respond to anti-VEGF therapy provided better results: BCVA improved significantly more (compared with baseline), and CMT decreased more in the early switch group compared with the late switch group [63]. In addition, no difference in the incidence of increased IOP was observed between the groups [63]. Comparable results were reported in Demir and colleagues’ study; the authors concluded that the central retinal thickness (CRT) decreased significantly more in the early switch group compared with the later switch group [64]. These results agree with those of a study by Ruiz-Medrano et al. [45]. Superior functional outcomes were observed in eyes with insufficient responses to anti-VEGFs in patients switched to DEX who had been receiving three monthly anti-VEGF injections compared with those who had been receiving more than three monthly anti-VEGF injections [65].

Cataract surgery can induce DME progression as well as the development of DME in patients with diabetes [28]. Several studies have reported improved functional and anatomic clinical outcomes in patients with DEX implants during cataract surgery [66,67,68]. Furino and colleagues conducted a study to evaluate functional and anatomical outcomes after combined phacoemulsification and intravitreal DEX implantation with standard phacoemulsification in diabetic patients with cataracts [69]. In the group with combined phacoemulsification and intravitreal DEX implantation, BCVA improved significantly more, and central subfoveal thickness decreased more [69]. Although this group had significantly higher IOP during follow-up at month 3 compared with baseline, IOP remained within the normal range [69].

Possibilities for future treatment include ziv-aflibercept, which was proposed as a new recombinant fusion protein and which has a mechanism of action similar to that of aflibercept; however, it is available at a lower cost than the proprietary anti-VEGF drug [70]. It was reported to be effective and safe in DME treatment and other retinal diseases; however, further studies are needed [70,71]. Because of the longer intravitreal half-life of the new generation anti-VEGF-A inhibitors, including brolucizumab, abicipar pegol, and angiopoietin combination drugs, improved prolonged edema reduction and less frequent injections appear to be required [11,28]. The preliminary results of studies currently in progress have suggested that anti-VEGF-A may have superior effectiveness compared with approved anti-VEGFs [11,28,72].

Rivera et al. reported evidence of reduction of DME through the consumption of lutein. In patients with ME who have lower levels of lutein, lutein consumption prevented and reduced possible complications [73].

A summary of the treatment options for DME is presented in Table 1.

### 3.2. Macular Edema Secondary to Retinal Vein Occlusion

Retinal vein occlusion (RVO) includes branch RVO (BRVO), central RVO (CRVO), and hemi-RVO, which are categorized according to the anatomic location of the occlusion [74]. In all hemorrhages and ME occur, leading to significant visual impairment [75].

Although LP has long been considered a primary treatment option, similar to DME, it has been replaced by other treatment methods. It was reported that although macular grid laser treatment reduced vision loss and the risk of vitreous hemorrhage in eyes with ME due to BRVO, it was ineffective against ME due to CRVO [15,74]. Zhang and colleagues additionally reported that LP cannot be performed in cases of retinal swelling with hemorrhage because the laser energy is absorbed and reduced; however, laser therapy may be used as rescue therapy for ME secondary to RVO [74].

Hayreh et al. has reported that in patients with ME due to RVO who respond poorly to anti-VEGF therapy or are incapable or reluctant to attend clinics for frequent anti-VEGF injections, grid laser treatment can be used combined with anti-VEGF therapy [76].

Intravitreal anti-VEGF injections are now considered the first-line treatment for ME associated with RVO, and their efficacy and superiority over other treatment methods have been demonstrated in many studies. Qian et al.’s meta-analysis reported that anti-VEGFs were the most effective therapy for ME secondary to both CRVO and BRVO [77]. The survey study, which was performed among retina specialists in Japan, revealed that anti-VEGF therapy was chosen as the first-line treatment for ME secondary to BRVO, and most specialists (82.4%) selected initial injection followed by a pro re nata (PRN) regimen; however, the opinions about the initiation and switching therapy varied between specialists [78]. As additional treatment in refractory cases, laser therapy was reported as the most common choice (35.9%), with 25.6% selecting vitrectomy, and 15.4% chosing to add steroid injections [78].

Anti-VEGFs used to treat ME due to RVO are similar to those used to treat DME; ranibizumab and aflibercept are used on label, whereas bevacizumab and conbercept have been used off label. Hykin and colleagues performed a prospective study to evaluate the effectiveness of ranibizumab, aflibercept, and bevacizumab for the management of ME due to CRVO [16]. They reported that mean changes in vision after 100 weeks of follow-up and treatment were not inferior with aflibercept than with ranibizumab; however, the mean number of injections given in the aflibercept group was lower than that in the ranibizumab group [16]. The mean changes in vision using bevacizumab compared with those using ranibizumab were similar, suggesting that the effectiveness of bevacizumab was neither equal nor superior to ranibizumab [16]. Conbercept is one of the newest anti-VEGFs and provided good treatment results in Chinese patients with RVO in a randomized clinical trial [79]. Xia and colleagues reported that conbercept significantly reduced retinal structural remodeling, inflammation, and oxidative stress in mice as well as in patients with ME due to RVO [75]. However, some patients with severe ME due to RVO did not experience significant benefit from conbercept [75]. The authors hypothesized that this may have been because conbercept only inhibits downstream VEGF inflammatory mediators and does not affect the upstream inflammatory mediators of VEGFs, such as PGE1, PGE2, and PGF2a [75]. Costa et al. reported that intravitreal anti-VEGF injections are prioritized over other treatment methods, including macular grid photocoagulation [80]. Compared with steroid injections, anti-VEGFs are superior because they have fewer side effects; as with their use in DME, steroids are associated with a higher incidence of increased IOP and cataract formation [80]. A systematic review and meta-analysis were performed by Liu and colleagues to evaluate the efficacy of conbercept and ranibizumab with or without LP in patients with ME secondary to RVO [81]. Both intravitreal conbercept and ranibizumab therapy with or without LP were effective in improving vision function in patients with ME secondary to RVO. The two anti-VEGFs did not differ significantly in BCVA improvement or adverse effects, and they resulted in similar visual gains [81]. However, conbercept reduced CMT more than ranibizumab with fewer injections [81]. Another systematic review performed by Spooner and colleagues evaluated 17 studies involving 1070 eyes [15]. It demonstrated that the management and outcomes of patients with CRVO varied greatly; however, anti-VEGF therapy significantly improved the anatomical and functional outcomes [15]. Although most eyes obtained a significant visual acuity gain, those treated with aflibercept and bevacizumab had significantly better outcomes than ranibizumab-treated eyes [15]. The incidence rates of ocular complications were low, including neovascular glaucoma (3.6%), vitreous hemorrhage (<1%), glaucoma (1.2%), and neovascular glaucoma (<1%) [15].

The management of cases refractory to anti-VEGF treatment is an ongoing dilemma, and therefore, the efficacy of steroids in patients with ME due to RVO has been explored in several studies. One study hypothesized that inflammation could be the first key mechanism to mechanical injury in RVO, and VEGF up-regulation may occur as a secondary effect of this inflammatory response [75]. Corticosteroids can significantly reduce inflammation, retinal vascular permeability, and the regulation of VEGF-A expression, and thus they have been used for the treatment of ME due to RVO [74]. The intravitreal dexamethasone implant is approved for the treatment of ME due to RVO [74]. Ming and colleagues performed a meta-analysis on the efficacy and safety of intravitreal DEX implants and anti-VEGFs for the treatment of ME due to RVO; the review included 4 randomized controlled trials and 12 real-world studies [19]. The authors reported that DEX implantation resulted in a comparable or smaller reduction in central subfield thickness (CST) at months 6 and 12 but introduced higher risks of elevated IOP and cataract induction [19]. It was concluded that compared with anti-VEGF agents, DEX implants required fewer injections but had inferior functional efficacy and safety [19].

The management of central and branch RVO and its long-term effects were evaluated in a 7-year follow-up study by Arrigo et al. performed in an Italian referral center [82]. Contrary to the previously discussed study, the authors reported that both CRVO and BRVO eyes exhibited significant visual acuity improvements secondary to intravitreal anti-VEGF or dexamethasone treatments and a significant reduction in CMT at the end of the follow-up. Furthermore, the authors highlighted a result that showed that the time at which the greatest improvement was observed differed between CRVO and BRVO; an earlier improvement was observed for CRVO (after 12 months of follow-up), and a later improvement was observed for BRVO (after 24 months of follow-up). However, after 2 years, both visual acuity and CMT remained stable until the end of follow-up.

Evidence of the value and importance of SML therapy in ME treatment is increasing. Buyru et al. compared the effects of intravitreal ranibizumab and SML treatment in two groups of patients with ME due to BRVO [83]. They concluded that the reduction in macular thickness and the increase in visual acuity were comparable for intravitreal ranibizumab and yellow SML treatment over 1 year. It was suggested that SML treatment may be useful in the treatment of ME due to BRVO. Eng and colleagues conducted a literature review on the efficacy of SML treatment for ME due to BRVO and reported that SML therapy resulted in a smaller reduction in ME compared with intravitreal anti-VEGF agents [84]. However, the authors concluded that SML treatment could be useful as adjuvant therapy with intravitreal anti-VEGF agents or steroids. Terashima et al. evaluated the efficacy of the combined therapy of intravitreal ranibizumab and 577 nm yellow laser SML photocoagulation for ME secondary to BRVO [85]. They concluded that combination therapy with intravitreal injections and SML was effective and decreased the frequency of intravitreal injections while maintaining good visual acuity. Similarly, a meta-analysis conducted by Chen et al. concluded that laser therapy combined with intravitreal ranibizumab injections had a strong effect, promoting its use for the treatment of ME secondary to BRVO in clinical practice [86].

Nanotechnology (nanocarriers) offers multiple benefits by promoting drug delivery across tissue barriers, controlling the release of a topically administered drug, improving bioavailability, and directing drugs to the target tissue [87]. An example of a nanosystem is the topical ophthalmic TA-loaded liposome formulation (TA-LF), which releases TA into the vitreous and retina [87]. It was reported to be safe and effective in rabbits as well as in patients with refractory pseudophakic cystoid ME. Navarro-Partida and colleagues evaluated its safety and efficacy in patients with ME secondary to BRVO who were given a topical instillation of one drop of TA-LF (TA 0.2%) six times a day for 12 weeks [87]. The results confirmed its effectiveness; a significant reduction in central foveal thickness and a significant improvement in BCVA were observed. No adverse events, including increased IOP, were reported. The authors suggested that as liposomes can function as nanocarriers of TA, they could allow topical ophthalmic therapy to become the primary treatment option instead of intravitreal drugs in patients with ME secondary to BRVO. Cheng et al.’s also showed that liposomes with TA in eye drops could be a new therapeutic approach for the effective treatment of retinal diseases [88].

Authors have investigated factors associated with the course of the disease and the response to the treatment. Kida and colleagues hypothesized that increased retinal venous pressure (RVP) plays an important role in the formation of macula edema; thus, they recently evaluated RVP before and 1 month after intravitreal ranibizumab injection to determine its effect on RVO-related ME [89]. They concluded that RVP decreased significantly after treatment; however, it remained significantly higher than the IOP. Rothman and colleagues assessed the impact of age on ME due to RVO and concluded that patients younger than 50 years old had higher baseline and final visual acuity, a lower incidence of cystoid macular edema at presentation, and received fewer intravitreal injections than older patients [90].

A summary of treatments for ME due to RVO is presented in Table 2.

## 4. Discussion

ME significantly reduces visual acuity independently of its cause. Long-standing ME is associated with irreversible visual impairment; thus, the management of this condition should not be delayed.

The resolution of DME is accompanied by macular atrophy due to permanent damage to the photoreceptors, and CST is not a reliable indicator of visual acuity, neither as a prognostic nor as a predictive factor of outcomes [91]. This highlights the importance of evaluating visual acuity as a functional outcome in studies evaluating the effects of ME treatment. Most of the studies reviewed evaluated both central retinal thickness and BCVA, determining its relevance.

Almost all studies comparing laser treatment with other methods of treatment noted that LP has not been the first-line treatment for DME and ME secondary to RVO for some time, as it has been replaced by more effective intravitreal anti-VEGF injections [27,28,29,77,78].

Although a lower incidence of complications was reported with SML treatment compared with conventional laser treatment, SML treatment has not shown superior effectiveness [31,32,33,34]. However, the use of a combined treatment may be an effective and safe alternative for ME treatment and may reduce the number of intravitreal anti-VEGF injections required [34,35].

Although some studies have reported superior efficacy of certain anti-VEGFs over others, the agents reported as superior vary. It is accepted that anti-VEGFs are typically chosen on the basis of baseline VA, drug price, and availability. The new generation of anti-VEGF-A inhibitors, including brolucizumab, abicipar pegol, and conbercept, are believed to be superior to the anti-VEGFs currently used in ME treatment because of their longer intravitreal half-life, higher potency, biochemical properties, and the reduced number of intravitreal injections required per unit time. However, extended studies and trials must be completed before the new drugs are approved [11].

Despite the overall efficacy of anti-VEGFs, many patients do not respond to them. It was reported that only 33–45% of DME patients on anti-VEGF agents showed three lines or more of visual improvement [28]. Forty percent of patients failed to achieve significant visual gains despite 6 months of intensive anti-VEGF therapy. ME persisted in 32% to 66% of eyes and usually affected visual acuity significantly [44].

Despite this, steroids are typically a second choice for both DME and ME due to RVO and are reserved for those who do not respond to anti-VEGF treatment. However, increasing evidence suggests an association between superior functional (increased BCVA) and anatomical (reduced CMT) outcomes and beginning steroid treatment earlier [52,61,63,64,65]. Although steroids are associated with increased IOP and cataract formation [80], this is not an inevitable outcome for all of the patients treated with steroids, as studies reported these side effects in less than half of patients. In addition, side effects could be caused not only by steroids but also by the preservatives used in their preparation [44]. Most of the studies reported a significant positive effect of intravitreal steroids in the treatment of ME, thus highlighting its advantage. The intravitreal FA implant is superior to the DEX implant because of its longer effect (up to 36 months); however, it is usually used to treat DME in patients who previously received a course of corticosteroids and did not experience a significant increase in eye pressure [11]. Furthermore, intravitreal FA was approved for DME, but it has not yet been approved for ME due to RVO. We did not identify any studies that compared DEX and FA in terms of effectiveness.

It would appear that, as of yet, a consensus on ME treatment has not been reached, particularly in cases that are resistant to standard treatment. We assume that artificial intelligence (AI) may be beneficial in addressing this issue. It was previously reported that AI was able to accurately predict posttreatment central foveal thickness and BCVA after anti-VEGF injections in DME patients; thus, it can be used to prospectively assess the efficacy of anti-VEGF therapy in DME patients [91]. The data regarding AI properties and possibilities in ME diagnosis and treatment prognosis are increasing [92]. Optical coherence tomography (OCT) is an indispensable tool for the application of AI as well as for determining the need for treatment and evaluating its effectiveness in patients with ME [93]. Until AI is widely and effectively incorporated in clinical practice, established imaging biomarkers may significantly contribute to DME management. Hyperreflective retinal foci (HRF) appear as intraretinal hyperreflective dots on OCT in patients with DME and are reported to be an important imaging marker of retinal inflammation [94]. Kim et al. suggested that patients with an increased number of HRF on OCT should be more frequently followed up for early intervention because they observed that a higher number of HRF on the spectral domain (SD) OCT was associated with early recurrence of DME after steroid implants [94]. It was also reported that the presence of subretinal fluid, the absence of HRF, and the integrity of the inner segment–outer segment layer could be OCT biomarkers for superior functional success [55]. Larger cysts (intraretinal cystoid spaces) are associated with poor visual prognosis, and the size of the cyst is correlated with the extent of macular ischemia [14]. An increased fundus autofluorescence (FAF) signal (hyper-autofluorescence) was associated with declining visual acuity and an increase in the macular thickness on OCT [95]. This highlights the properties of FAF as an additional tool that may help monitor the progression of DME and its response to treatment.

## 5. Perspectives

New therapies, including anti-VEGF-A inhibitors (brolucizumab and abicipar pegol), are under investigation and may be more effective in ME treatment compared with previous anti-VEGFs [11]. A suprachoroidal TA delivery system in DME patients has been investigated as well, and the preliminary results are promising [96]. Nanotechnology was reported to be safe and beneficial in its ability to ensure TA delivery to the retina using topical drops.

The SML is absorbed by xanthophyll pigment, allowing for treatment close to the fovea [84]. It can promote the absorption of edema, hemorrhage, and exudation, and it can improve the retinal oxygen supply and reduce vascular permeability [86]. This relatively new laser technique is superior to a conventional laser because it does not cause structural damage to the retina. Although SML therapy has not shown superiority when used alone in ME treatment, in most of the reviewed studies, SML therapy was reported to be an effective additional treatment method when combined with intravitreal anti-VEGF injections in the treatment of DME and ME due to RVO. Both methods have some limitations and possible complications; however, when combined, they not only effectively reduce ME and increase VA but also reduce the number of intravitreal injections. Therefore, this combined treatment could lower healthcare costs and the burden on patients by reducing the frequency of clinic visits.

With the emerging era of AI, this technology may soon be beneficial in selecting the most effective and appropriate treatment in patients with ME. Promising results were reported in a recent study performed by Gallardo and colleagues [97]. They used machine learning classifiers to predict low and high anti-VEGF treatment demands for patients with DME, RVO, and neovascular age-related macular degeneration treated according to a treat-and-extend regimen. The authors highlighted the ability to predict the low and high treatment demands in all groups of patients with similar accuracy, along with the capability to predict low demand at the first visit before the first injection. Further research is needed to establish the individual treatment demands for patients and consolidate the properties of AI in clinical practice.

## Figures and Tables

**Table 1 jcm-10-05300-t001:** Summary of treatment of diabetic macular edema.

	Considered First-Line Treatment	Insufficient Response to Anti-VEGF	
**DME**	**Intravitreal anti-VEGF injections** ○ Bevacizumab, ranibizumab, and aflibercept are the most used anti-VEGFs, and many studies have not identified significant differences in outcomes between them ○ The choice of one anti-VEGF over another depends on baseline BCVA	**Intravitreal steroid (DEX/FA) implants** ○ Sustained drug release for a specific period ○ Acts on different targets than anti-VEGF agents by reducing inflammation, decreasing the disruption of the blood–retinal barrier, and interfering with retinal angiogenesis ○ A slow-release 700 µg dexamethasone intravitreal implant can be considered as an option for first-line treatment in center-involving DME ○ DEX can be considered as first-line therapy in pseudophakic patients without advanced or uncontrolled glaucoma ○ FA can be considered in pseudophakic patients in whom DEX has been well-tolerated ○ * TA—has not been approved for DME	**Micropulse laser therapy/** **conventional focal laser therapy** ○ Helps improve or stabilize visual function and decrease the macular thickness ○ Can reduce the number of intravitreal injections when used as a combined treatment

DEX—dexamethasone, DME—diabetic macular edema, BCVA—best corrected visual acuity, FA—fluocinolone acetonide, VEGF—vascular endothelial growth factor, TA—triamcinolone acetonide, * not an approved treatment.

**Table 2 jcm-10-05300-t002:** Summary of treatments for ME associated with RVO.

ME associated with RVO	First-line treatment	Cases resistant to anti-VEGF
	○ Intravitreal anti-VEGF injections ○ The superiority of agents in studies varies ○ Anti-VEGFs are chosen on the basis of baseline VA, drug price, and availability	○ Intravitreal steroid (DEX) implant ○ SML/conventional focal laser therapy as combined therapy

DEX—dexamethasone, ME—macular edema, RVO—retinal vein occlusion, SML—subthreshold micropulse laser, VA—visual acuity, VEGF—vascular endothelial growth factor.

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
