# Peer review of "Treatment of Macular Edema in Vascular Retinal Diseases: A 2021 Update"

_jcm, 2021, doi:10.3390/jcm10225300_

Round 1

Reviewer 1 Report

REVIEWER’S COMMENTS

The JCM manuscript Cystoid macular edema treatment. An update 2021.by Grzybowski et al reviews the types and treatment regimens for macular edema.

Please revise this manuscript per the comments below:

  1. Apart from the diseases/ conditions mentioned in this review article, macular edema also results from other disease(s)/ condition(s) such as AMD (Age-related Macular Degeneration) and Vitreomacular traction. Many people develop macular edema after eye surgery as well. Please discuss these additional causes in this review article.
  2. Please discuss briefly the epidemiology and underlying genetic causes of each of the diseases/ conditions being reviewed in this article.
  3. Please organize the major sections into sub-sections such as: causes, mechanism, epidemiology, and treatment, so that it is easier to follow.
  4. Please discuss briefly the nutritional supplements that are known to help with alleviating macular edema.
  5. Please be consistent with the style of references.

Author Response

  1. Other conditions have been included (page 2-4)
  2. Information about the epidemiology and genetics included in the page 3-4
  3. Text was divided into sections , subsections
  4. The information which was found about the supplements for the discussed conditions - end of page 10

Reviewer 2 Report

1) The manuscript requires major editing/revision of English language to improve readability and clarity.  

2) A bried discussion of other major causes of macular edema in the introduction section is recommended; for example: post-operative macular edema, macular edema secondary to vitreoretinal interface disease, macular edeema secondary to side effects of topical or systemic drugs, macular edema secondary to inherited retinal diseases, etc. 

3) There should be tables to organize/list the publications evaluated in this review and a summary of their findings/recommendations 

4) It would be important to discuss the clinical practice pattern of treatment regimen diabetic macular edema or macular edema secondary to reitnal vein occlusion: treat and extended vs PRN. 

5) It would be helpful to refer to major clinical trials/studies by their commonly recognized. For macular edema secondary to retinal vein occlusion: CVOS, BVOS, CRUIS BRAVO, SCORE, SCORE2 GALILEO, COPERNICUS, VIBRANT, etc. For diabetc macular edema: ETDRS, RISE, RIDE, DCR protocol, DA VINCI Study, Bolt study, MEAD study, DRCR Protocol V, etc

6) Might be better to just focus on macular edema seconadry to retinal vascular diseases (diabetes, retinal vein occlusion), and leave macular edema secondary to uveitis for another paper. 

Author Response

  1. The English language of manuscript was reviewed and edited
  2. Other conditions of ME included and discussed in the introduction (page 2-3)

4. Intravitreal injections treatment regimens discussed in page 4 (dor DME),in page 14 (for RVO)

6. Focused on the tretament of DME and ME due to RVO

Reviewer 3 Report

The authors present a very interesting and well summarized review of the current treatment for diabetic macular edema and edema associated with ocular vein occlusion and uveitis. I have few reservations that should be addressed.

P.3 L.85 “Based on European Society of Retina Specialists (EURETINA) guidelines published in 2017 focal or grid laser should be utilized mainly for non-center involving DME [11].”

Authors should mention that these guidelines stated that “Relative indications include laser application especially to the vasogenic subform of DME, which is clinically characterized by the presence of focally grouped MA and leaking capillaries”.

Moreover, if grid laser is not further recommended because of retinal scars, focal laser targeting capillary macroaneurysms are beneficial as a second line treatment (1,2) and maybe a first line treatment as a combined treatment to reduce the number of antiVEGF injections. Authors should mention this hypothesis and the randomized controlled trial that is currently evaluating it(3).

P.5 L.160 “However, the TA frequently increases the risk of intraocular pressure elevation, thus it is used less frequently compared to its alternatives [25].”

Authors should also mention the risk of pseudoendophthalmitis(4,5) and retinal toxicity (6–8) associated with intravitreal triamcinolone acetonide.

P.7 L.206 : “Baker and 206 colleagues found that for patients with DME and excellent visual acuity (defined as 20/25 or better), observation appeared to be a non-inferior initial management strategy compared to intravitreal aflibercept or laser 208 photocoagulation in terms of visual acuity outcomes at 2 years [42]”

Authors should indicate here that the study of Baker et al. also showed that initial focal/grid laser significantly reduces the risk of aflibercept injection during follow-up.

P.8 TABLE 1

As some readers may not read the entire text, Table 1 should be more precise and readable and cover some “precautions for use”

As the authors stated in the text, triamcinolone acetonide has not been approved for DME. So, this information should appear in Table 1 and triamcinolone acetonide should not be presented at the same level as DEX and FA.

Moreover, it should be written that

  • DEX can be considered as a first-line therapy in pseudophakic patients without advanced or uncontrolled glaucoma(9)
  • FA can be considered in pseudophakic patients in which DEX has been well tolerated

There are still some relative indications for conventional focal laser therapy and SML has not proved its superiority nor non-inferiority versus conventional focal laser therapy. So, conventional focal laser therapy should appear in Table 1 with SML as possible additional treatments.

P.9 L.257 “however, laser therapy may be used as rescue therapy for a ME secondary to RVO”.

Authors should also recommend the comprehensive review of M.Hayreh “Photocoagulation for retinal vein occlusion” published in 2021, in which he concluded “in eyes with macular edema due to RVO that respond poorly to anti-VEGF therapy or are incapable or reluctant to come for frequent for anti-VEGF injections, grid laser can be used combined with anti-VEGF therapy.”

P.12 Table 2

Conventional focal laser therapy should appear in Table 2 with SML as possible additional treatment.

  1. Karti O, Ipek SC, Saatci AO. Multimodal Imaging Characteristics of a Large Retinal Capillary Macroaneurysm in an Eye With Severe Diabetic Macular Edema: A Case Presentation and Literature Review. Med Hypothesis Discov Innov Ophthalmol. 2020;9(1):33‑7.
  2. Castro Farías D, Matsui Serrano R, Bianchi Gancharov J, de Dios Cuadras U, Sahel J, Graue Wiechers F, et al. Indocyanine green angiography for identifying telangiectatic capillaries in diabetic macular oedema. Br J Ophthalmol. avr 2020;104(4):509‑13.
  3. Paques M, Philippakis E, Bonnet C, Falah S, Ayello-Scheer S, Zwillinger S, et al. Indocyanine-green-guided targeted laser photocoagulation of capillary macroaneurysms in macular oedema: a pilot study. Br J Ophthalmol. févr 2017;101(2):170‑4.
  4. Fung AT, Tran T, Lim LL, Samarawickrama C, Arnold J, Gillies M, et al. Local delivery of corticosteroids in clinical ophthalmology: A review. Clin Experiment Ophthalmol. avr 2020;48(3):366‑401.
  5. Mason RH, Ballios BG, Yan P. Noninfectious endophthalmitis following intravitreal triamcinolone acetonide: clinical case and literature review. Can J Ophthalmol J Can Ophtalmol. déc 2020;55(6):471‑9.
  6. Lang Y, Zemel E, Miller B, Perlman I. Retinal toxicity of intravitreal kenalog in albino rabbits. Retina Phila Pa. août 2007;27(6):778‑88.
  7. Schulze-Döbold C, Weber M. Loss of visual function after repeated intravitreal injections of triamcinolone acetonide in refractory uveitic macular oedema. Int Ophthalmol. oct 2009;29(5):427‑9.
  8. Arndt C, Meunier I, Rebollo O, Martinenq C, Hamel C, Hattenbach L-O. Electrophysiological retinal pigment epithelium changes observed with indocyanine green, trypan blue and triamcinolone. Ophthalmic Res. 2010;44(1):17‑23.
  9. Kodjikian L, Bellocq D, Bandello F, Loewenstein A, Chakravarthy U, Koh A, et al. First-line treatment algorithm and guidelines in center-involving diabetic macular edema. Eur J Ophthalmol. nov 2019;29(6):573‑84.

Author Response

Thank You for the suggested information and references, they were incorporated in the manuscript, following:

  1. Relative indications include laser application especially to the vasogenic subform of DME, which is clinically characterized by the presence of focally grouped MA and leaking capillaries”. Page 6, text in red
  2. the risk of pseudoendophthalmitis(4,5) and retinal toxicity (6–8) associated with intravitreal triamcinolone acetonidePage 8, text in red
  3. Baker et al. also showed that initial focal/grid laser significantly reduces the risk of aflibercept injection during follow-up Page 9, text in red
  4. Supplemented information in Table 1 - page 11
  5. “in eyes with macular edema due to RVO that respond poorly to anti-VEGF therapy or are incapable or reluctant to come for frequent for anti-VEGF injections, grid laser can be used combined with anti-VEGF therapy Page 12, text in red
  6. table 2 - page 15